# Visiting with Elders—Aging, Caregiving, and Planning for Future Generations of American Indians and Alaska Natives

**Cole Allick** [1,*] **and Marija Bogic** [2]

1 Department of Indigenous Health, University of North Dakota, Grand Forks, ND 58202, USA
2 Institute for Research and Education to Advance Community Health (IREACH), Elson S. Floyd College of Medicine, Washington State University, Seattle, WA 98101, USA; marija.bogic@wsu.edu
* Correspondence: cole.c.allick@und.edu; Tel.: +1-(701)-550-1936

**Abstract:** (1) Background: To address the importance of engaging American Indian and Alaska Native Elders in a dialogue about healthy aging and fill the gap in the scholarly literature on this topic. (2) Methods: This study conducted a listening session with Elders who attended the 2021 National Indian Council on Aging (NICOA) Annual Conference in Reno, Nevada. The listening session was audio-recorded and transcribed for thematic analysis by two analysts. (3) Results: Important insights regarding American Indian and Alaska Native Elders' perspectives on planning for future care and aging-related diseases, such as Alzheimer's disease. (4) Conclusions: This study is one of the first to engage American Indian and Alaska Native Elders in a conversation about health aging. Calls for intergenerational solidarity, protection of Elders, education, and relationality were found to be important themes.

**Keywords:** American Indian; Alaska Native; aging; older adults; Alzheimer's disease; long-term support and services

## 1. Introduction

The American Indian and Alaska Native (AI/AN) population aged 65 and older is rapidly growing. By 2050, this age cohort is expected to triple to 1.6 million, with those aged 85 years and older predicted to increase sevenfold to 0.3 million (Hebert et al. 2013). These people are often referred to as Elders, with individual Tribal Nations defining their eligibility (Centers for Medicare & Medicaid Services 2015). From an age standpoint, these individuals are often 55 and older. Elders are more than just a fast-growing segment of Tribal Nations; they are also bearers of generational knowledge and carry on the resiliency of their people through teachings, stories, and guidance (Garrett et al. 2014). As cultural leaders, respecting Elders is often seen as a way to honor historical practices and important traditions of community. These traditions include Elders' social participation and sharing of their knowledge, values, and culture, which play a crucial role in enhancing individual and community wellness (Viscogliosi et al. 2020).

The growing number of Elders draws attention to the need to better understand aging-related diseases, such as Alzheimer's disease and related dementia (ADRD) and its precursor, mild cognitive impairment (MCI) (Knopman and Petersen 2014). AI/AN populations are the most underrepresented racial minorities in ADRD research, and Indigenous researchers and stakeholders have called for more research to address this gap and for the incorporation of Indigenous knowledge and experiences (Browne et al. 2017; Garrett et al. 2015; Jervis and Manson 2002; Mehta and Yeo 2016). This call directly responds to the fact that much of the research has been conducted on Indigenous communities rather than with them (Israel et al. 2010; Morton et al. 2013). These types of projects often have little or no input from communities, leading to communities not receiving any answers or benefits from the study or its findings. Given the importance of Elders in AI/AN communities, their contribution to individual and community wellness, and their expertise in the topic,

it is crucial to involve the Elder community in the research process. By including Elders, research will be conducted more effectively, ensuring community benefit and adoption of findings.

The few existing studies conducted within AI/AN communities have identified a lack of understanding and general awareness of ADRD, as well as inadequate resources, particularly those tailored to meet the cultural needs of these communities (Griffin-Pierce et al. 2008; Hulko et al. 2010). For example, a study conducted in Alaska Native communities interviewed caregivers and providers and found a lack of understanding, resources, and general awareness of ADRD in these communities (Lewis et al. 2021). Similarly, another study focusing on ADRD knowledge among AI/AN people in the Pacific Northwest revealed that culturally informed materials are still needed in order to address knowledge gaps concerning ADRD (Jernigan et al. 2020). Furthermore, research on the apolipoprotein E (APOE) ε4 allele, which confers a higher risk of neurodegeneration and Alzheimer's disease (AD), has found no evidence of that risk in American Indian people, with calls to evaluate potential protective factors that may be present (Suchy-Dicey et al. 2022). A recent systematic review of ADRD risk factors in Indigenous populations concluded that the tendency to attribute disproportionately high rates of ADRD in Indigenous populations to genetic factors instead of socioeconomic factors can distract from the importance of modifiable risk factors, which greatly outnumber the non-modifiable factors (Walker et al. 2020). This demonstrates the crucial role of socioeconomic context in the pathogenesis of ADRD in Indigenous populations. AI/AN individuals experience a disproportionate burden of ADRD risk factors (Barnes et al. 2010; Indian Health Service 2020), including hypertension (Breathett et al. 2020), type 2 diabetes (Athanasaki et al. 2022), smoking (Rusanen et al. 2011), and low socioeconomic status (Marden et al. 2017). Elders often face poverty, geographic isolation, and limited access to health care and other health services, such as long-term supports and services (LTSS) (Centers for Medicare & Medicaid Services 2016). LTSS refers to a broad set of services that help people with personal or health care needs in addition to activities required by daily life over a prolonged period of time (Centers for Medicare & Medicaid Services 2015). The current infrastructure for providing LTSS for AI/AN Elders is minimal, and national efforts to quantify the need for these services often exclude Tribal communities (Centers for Medicare & Medicaid Services 2016).

While early scholars have attempted to quantify ADRD in AI/AN communities and have even engaged in some early qualitative work to understand knowledge, few if any have actively engaged Elders in this discussion. This study sought to fill this gap in research on ADRD in AI/AN communities by hosting a community listening session with Elders to learn about their perceptions of healthy aging and their needs, as the number of AI/AN Elders is expected to significantly increase over the next three decades.

## 2. Results

### 2.1. Demographics

Table 1 highlights the demographic information of the participants, who were largely rural or reservation-based (69%) and were from Tribes in Arizona, California, Nebraska, New Mexico, Oklahoma, Oregon, South Dakota, and Washington. Despite not having complete demographic data on all participants, the data indicated a strong presence of Elders in the sample, coinciding with the unique nature of the conference in which this work was conducted. Additional data showed that at least two participants were part of the Tribal Council or Elder Council for their community, which is important to acknowledge as these are key leaders in Tribal communities.

**Table 1.** Demographics of AI/AN participants.

| | | |
|---|---|---|
| Total respondents, *n* | | 39 |
| Female, *n* (%) | | 33 (84.6%) |
| Age, *M* (*SD*) | | 66.3 (9.3) |
| Some college/college degree, *n* (%) | | 28 (92.3%) |
| Personal doctor/health care provider, *n* (%) | | 27 (69.2%) |
| Location, *n* (%) | Reservation | 15 (38.5%) |
| | Rural not reservation | 12 (30.8%) |
| | Large metropolitan | 5 (12.8%) |
| Marital status, *n* (%) | Never married | 2 (5.1%) |
| | Married/partnered | 12 (30.8%) |
| | Divorced/separated | 9 (23.1%) |
| | Widowed | 14 (35.9%) |
| Employment status, *n* (%) | Full-time | 15 (38.5%) |
| | Part-time | 4 (10.3%) |
| | Retired | 14 (35.9%) |

Note: One respondent chose not to complete all of the demographic questions.

## *2.2. Themes*

The data revealed six themes, detailed below, concerning the Elders' perceptions of healthy aging and their needs. Within each theme, quotes are shared that the co-authors agreed were the best fit for that theme; however, in the analysis phase, it was clear that there were overlapping themes and concepts embedded within many of the quotes. The themes, codes, and quotes are also noted in Table 2.

**Table 2.** Codebook.

| Main Theme | Definition & Codes | Example |
|---|---|---|
| Intergenerational trauma | Any instance that Elders mention boarding schools or mentions of personal/community histories. E.g., distrust of Western government, apprehension, forgo services | *"You know, like from boarding schools and that, you know, a lot of, a lot of people were... A lot of Elders were raised, raised that way to stay quiet. So, behavioral health is real important."* *"They took it as trying to remove them out of their home, shoved them somewhere and be forgotten."* |
| Intergenerational solidarity | Any mention of the important role Elders play as leaders and keepers of Traditional Knowledge E.g., relationship between youth an Elders, cultural importance, leadership, role models, wisdom, knowledge transfer | *"But that's what our generation is losing is that cultural spirituality, the love of one another, uh, physically, mentally and spiritually."* *"...was that when you're raised with your Elders, you have that respect, you learn your treatment of how you treat your Elders."* |
| Protection of Elders | Any mention of respect for Elders and treating them appropriately E.g., neglect, physical harm, disrespect, exploitation, isolation, separation | *"What we're finding out is that a lot of times they're not being treated, um, like they should be treated, they forget about them or, you know, they don't take care of them properly and stuff like that. We get complaints."* |

**Table 2.** *Cont.*

| Main Theme | Definition & Codes | Example |
|---|---|---|
| Relation and connection | Any mention of social connection with Elders E.g., community engagement, volunteering, employment, reducing loneliness | *"People right there at home, their family could come see 'em. They could bring 'em meals and then they would have someone always able to come see 'em. When I see 'em, they say, 'Just come look at me. I'm so lonesome, just come look at me. Talk to me for 10 minutes.' They're lonesome."* |
| Services for Elders | Any mention of services needed or provided for Elders E.g., long-term supports and services, both Tribal and non-Tribal, funding | *"And so, I would just like to see, you know, all Native Americans if they come into IHS, you know, they're getting that care, they're getting, well, transportation, making sure that they get their meals, making sure that, um, someone's checking on 'em."* |
| Education for community | Any mention of education needed for people to understand ADRD E.g., stigma, fear, lack of knowledge about ADRD or services | *"And it took me to educate myself to know to have the patience for him, and to just go with what he was saying and not try to correct him. Like I was saying before, I think that's the biggest thing is, is educating."* |

### 2.2.1. Intergenerational Trauma

Participants underscored the importance of recognizing the historical experiences of Elders that have resulted in distrust or apprehension toward Western systems. Participants mentioned experiences such as boarding schools and forced removal from home communities. Community-level apprehensions combined with previous traumatic experience with health care leads to fears among Elders of facing an ADRD diagnosis and being forced into LTSS unwillingly.

- "'Cause they're afraid to call and say anything 'cause, you know, we're raised in our own way, you know, to be quiet, don't say nothing, you know, and that's just... You know, like from boarding schools and that, you know, a lot of, a lot of people were... A lot of Elders were raised, raised that way to stay quiet."

### 2.2.2. Intergenerational Solidarity

Participants thought of Elders as important keepers of knowledge and traditions in community. Throughout the conversation, participants reiterated the need for youth and Elders to restore and deepen their bonds. They called on communities to advocate for knowledge dissemination by returning to traditional ways of knowing and being, where Elders are seen as wisdom keepers and role models. They also called for everyone to sit and acknowledge Elders with respect, compassion, and openness to their experiences.

- "The need for cultural and spiritual practices to be maintained in order to bridge the generational gap and ensure that Elders are respected."

### 2.2.3. Protection of Elders

Participants indicated that many Elders are being abused, disrespected, or exploited in their communities with complaints often filed. These Elders may also be isolated or separated from their families or communities. Some participants specifically mentioned instances where they have visited Elders in long-term support facilities where they are not being treated appropriately. Other participants called out the exploitation of Elders for financial gain at the expense of the Elders' own personal care.

- "Taking money from the Elders, taking... Not, not providing for their care, stuff like that."

### 2.2.4. Relation and Connection

Participants shared many stories of Elders' expressing loneliness and yearning for social connection. They indicated that these Elders feel forgotten or that the community, especially the youth, no longer carry the same respect, admiration, and concern for Elders. Participants indicated that these Elders are fearful of being sent to an assisted living facility, where they may face isolation from their loved ones and fellow community members. Participants mentioned that in a Tribal sense, connections and relations often include more than just blood relatives. Participants shared stories and instances of how they took it upon themselves to send things such as birthday cards to Elders in their community in order for them to feel connected.

- "...the social, they just want more community opportunities to socialize together and talk and share and, you know, be part of what's happening with each other and supporting each other, doesn't always occur."

### 2.2.5. Services for Elders

Throughout the listening session, participants presented multiple instances of system-level barriers and opportunities. Participants shared that many communities offer some services for Elders, yet they are often underfunded or are awaiting approval from the federal government and/or Indian Health Service (IHS) in order to be implemented. Participants indicated that the community understands the need for more comprehensive and culturally informed services for Elders; however, many noted that these services can be difficult to implement since they are often hindered by systemic issues, such as jurisdiction.

- "My concern is what's going to happen to me in 10 years as my, my failure. I don't want to go to a nursing home. So I'll start advocating to our tribe to please do some kind of more senior housing, more benefits for care, uh, you know, as we all age."

### 2.2.6. Education for Community

Participants mentioned a need and desire for education so that people may better understand dementia, including ADRD. They specifically pointed out a stigma and lack of knowledge about ADRD among their communities, with many sharing direct, personal experiences with Elders in their lives. Many participants mentioned patience as a primary issue among community members in supporting Elders in their life, who may be suffering from ADRD.

- "So, I think for me, if there was more information sharing or even community forums or people discussing things. There were just so... There were little things happening towards end of life that I didn't realize or pick up on, or maybe I even ignored because I was so respectful of them that had I realized what was going on, I could have helped in a better way. So then afterwards, when I go back to remembering the incident, when I feel guilty thinking, 'Oh, I could've done better. Or what if I'd have known that I could have did this. I could have changed their comfort level.' That sort of thing, even though I knew I did the best with what I knew. I think having that information could just be such a difference when you're in a caregiving situation."

## 3. Materials and Methods

### 3.1. Researchers and Reflexivity Description

The lead author is a citizen of the Turtle Mountain Band of Chippewa Indians. He attended school near his Tribal Nation and carries a multi-pronged identity of a rural, Tribal citizen who has since moved to a large, metropolitan area. His spirit name, Miingun Waaju (Mountain with Wolf Spirit), connects him to his land, family, and community. He shares his name to provide a connection to the readers and to ground him in this work that was completed on Indigenous lands in Seattle, Washington, where he resides as a guest. This work is deeply rooted in the teachings of his spirit name, which carry lessons about balance, practicality, and how to use his gifts for the benefit of all communities. He shares this

introduction with the inclusion of his spirit name so that he may share this work and honor the expectations associated with his spirit name throughout his career.

The co-author of this paper is an older, white, eastern European, female refugee and immigrant, who works with Indigenous communities. She has been displaced from her homeland for more than 30 years and is aware of the privileges and experiences that come with her position as a white, older female researcher. She is conscious of how her perspectives and experiences may differ from those of the research participants and is aware of her own subjectivities and the potential power dynamics of her role in relation to the lead author, an American Indian person. To ensure ethical and culturally respective research, the researcher has reflected on her positionalities and privileges and is committed to being reflexive throughout the research process.

### 3.2. Interview Format and Content

The community listening session was chosen in order to facilitate dynamic consideration of ideas and honor and encourage diverse perspectives, particularly those of traditionally marginalized and underserved people (Ardoin et al. 2022; Rodriguez et al. 2011). This community-driven approach has been identified as particularly beneficial in fostering dialogical reflection, interaction, and discussion on a topic about which participants are experts (Ardoin et al. 2022). During the single listening session, seven open-ended questions were asked in order to gain insights into cultural understandings of ADRD among AI/AN Elders. The lead author of this research, who is employed at Washington State University's Institute for Research and Education to Advance Community Health, engaged the individual who leads the Institute's Methods Core to support the development of a listening session guide, including the open-ended questions. The individual was chosen as they have expertise in question and protocol development, as well as experience conducting work alongside Tribal communities. The questions were developed based on the existing literature gap and focused on the role of Elders, availability of LTSS, caregiving, and caregiver support. These questions are listed in Table 3.

**Table 3.** Listening session questions.

| Number | Question |
|---|---|
| 1 | What role do Elders play in your community? |
| 2 | Are there services in your community to support Elders? If so, what are they? |
| 3 | Are there services or activities geared towards Elders that are missing from your community? If so, what are they? |
| 4 | In your community, is there availability of long-term care services (assisted living, senior living communities, in-home care) that allow Elders to remain at home or close to home? Are they Tribal or non-Tribal? |
| 5 | Whose role is it to understand and care for the rapidly growing Elder population? |
| 6 | Is anyone here a caregiver or has been a caregiver of someone experiencing cognitive decline, dementia, or Alzheimer's? If so, what was your experience? |
| 7 | Does your community have services to support caregivers providing care for someone with cognitive decline, dementia, or Alzheimer's? |

### 3.3. Participants, Setting, and Procedure

The sample was recruited from a convenience sample of AI/AN people attending the 2021 National Indian Council on Aging (NICOA) biannual conference, which occurred from August 1st through August 6th in Reno, Nevada. For nearly 50 years, NICOA has served as the nation's foremost advocate for AI/AN Elders. NICOA membership is exclusive only to AI/AN people aged 18 years and older, though the vast majority are Elders (55 and older). Thus, the inclusion criteria were (a) self-identified American Indian and/or Alaska Native and (b) participants aged 18 or over. While this project is focused on working with

Elders, those that do not meet the age criteria for Elders were also included to reflect the perspectives of all community members. Leadership at NICOA invited the lead author to present about his experience with the PhD program in Indigenous Health at the University of North Dakota and interest in ADRD among AI/AN communities, prompting the creation of this project.

The listening session was presented to attendees on 2 August 2021, via a 90 min breakout session, titled "Supporting an Indigenous Health Ph.D. Student Focused on Healthy Aging", in a room with 75-person capacity. Per the request of NICOA leadership, the session started with a short presentation of the University of North Dakota's Indigenous Health PhD program, followed by an introduction to the lead author (Cole Allick), and led directly into a listening session with attendees. The listening session was conducted in a room that was comfortable and private. As part of the listening session protocol, participants went through a consent process that included information about the topics discussed, how to participate, recommendations to maintain confidentiality, and to not repeat what was shared. Participants were also informed that they could leave the room at any point and were allowed to not participate in any questions they found uncomfortable. After introducing the research project and explaining the purpose of the session, participants were asked the seven open-ended questions.

Given the format of the listening session, in which participants were allowed to enter and leave as they desired, the exact number of participants is difficult to articulate. However, the room was at capacity for the duration of the listening session, leading to a reasonable estimate of approximately 70 participants. Further, a separate survey was provided for the first 50 participants, which resulted in demographic information for a subset of listening session attendees.

All recruitment, data collection, and analytic procedures were approved by the Washington State University and University of North Dakota Institutional Review Boards.

*3.4. Data Analysis*

The listening session was audio-recorded and transcribed using Rev Digital Transcription Services. Transcribed data were independently analyzed by two analysts—the lead author and a co-analyst unrelated to the study. The lead author chose a co-analyst to mitigate any bias, given his identity as a Tribal citizen, and to balance the final product to be useful for all audiences. In addition, the inclusion of two independent analysts is best practice for this type of data analysis. After securing a transcript, the analysts conducted a thematic analysis using the six-stage approach outlined by (Braun and Clarke 2006). The first stage of analysis involved a familiarization with the data, which was undertaken by closely reading the transcript. In the next stage, initial coding was conducted to identify meaningful units of data. This was followed by the development of a coding framework to categorize the data. The coding framework was created through inductive coding, which follows a ground-up approach of deriving codes from the data collected. After independently coding the transcript across questions, the two analysts compared codes, and substantive coding differences were resolved through discussions. Next, the data were organized into themes, which were then refined into a set of core themes. The final two stages involved the interpretation of the data and the writing of the results.

**4. Discussion**

To our knowledge, this is one of the few projects that has actively engaged Elders in a direct conversation about planning for their futures. In doing so, one message rings clear: we must acknowledge intergenerational trauma and resiliency with equal vigor. The use of intergenerational solidarity provides a strong call to action while concurrently uplifting Tribal Nations, their sovereignty, the cultural importance of Elders, and the rich histories in these communities. The use of this term allows for a deeper connection to each of the themes identified in this project, as it underscores the importance of each theme and provides a focus on tangible, pragmatic changes that can be advocated for. This discussion

illuminated changes necessary at all components of the socio-ecological model, including the individual, relationship, community, and societal levels.

First, at the individual level, there is a need for everyone to be grounded in their own identities regardless of their status as a Native or non-Native person, as demonstrated by the authors in their reflexivity statements. This is an important first step, as each of us center ourselves and undertake this work to serve Elders. This may be especially acceptable in a Tribal sense, as Elders feel connected to those that they may or may not share a biological relation to. The participants indicated a disconnect between community members and Elders due to a loss of respect for tradition, leaving many Elders feeling "forgotten." Through reflection and positionality, those who choose to pursue this work can address the theme of intergenerational solidarity to connect with Elders and begin to focus on the relationship level of the socio-ecological model. These relationships need to make space for understanding the Elders' histories, including their resiliency. Integrating the individual and relationship levels of the socio-ecological model would ensure that any advocacy for the Elders can be thorough and comprehensive.

The community level presents many opportunities to address the concerns illuminated by the participants. Communities need more education and awareness about ADRD and have indicated a desire to learn. This includes Elders, as they too are affected by the stigma and lack of knowledge about ADRD. Additional appeals for the protection of Elders include a call for Tribal community members to watch out for Elders who may be experiencing abuse or exploitation in their community. Many participants articulated this as a focus; however, explicit calls to action at the community level are necessary to ensure that Elders are supported and protected in their own communities. Another opportunity for community-driven ways of staying connected to Elders across the community is through activities and engagement geared toward addressing loneliness. One example is to have youth spend more time with Elders and reiterating that connection with Elders can come in many forms, such as sending an Elder a letter or birthday greeting even if you are not directly related to them. Another is to ask language keepers to share their knowledge in schools and other community settings. Concurrently, Elders remain active members of their communities, with many still caregiving for their grandchildren, volunteering at community events, teaching language or traditions to community members, or maintaining a role in Tribal government. These Elders use this purpose and service as a way to remain connected to their community and combat the loneliness that has been identified as a concern for aging populations.

At the societal level there is a clear lack of funding and culturally informed LTSS for Elders. This must be addressed by community champions and leaders, such as those who attended this listening session. The inclusion of Tribal Leaders, such as Tribal or Elder Council members, allows for the issues identified in this conversation to be advocated for through means afforded to Tribal Nations as they exert their sovereignty. NICOA represents a strong partnership for these potential solutions, as its mission is to serve Elders across Indian Country. Beyond callouts for more services for Elders, NICOA can also advocate alongside Tribal Nations for the protection of Elders through law and policy. NICOA continues to be a strong proponent for projects such as this one, which aim to increase our knowledge of ADRD and support the growing number of Elders across Indian Country. Relationships such as these are vital for addressing many of the systematic barriers faced by Elders in Indian Country.

## 5. Implications

This study provides critical insights for future, larger-scale efforts to identify policy, practice, and additional research that can address the concerns articulated by Elders in this analysis. The findings illustrate that Elders maintain many systems and relationships that influence their perceptions of aging and aging-related diseases, such as ADRD. The insights and context shared in this dialogue need to be considered when creating ways to better serve this rapidly aging community over the next three decades. Intergenerational

solidarity is required among all community members committed to caring for Elders, as it takes an active, strength-based approach to focus on the myriad systems that serve Elders. Partnerships with Tribal Nations, stakeholders, and external partners such as NICOA are paramount to the success of these endeavors and should be actively sought out. The findings of this study have been shared with the members of NICOA via the 2023 Annual Conference to provide benefit, respect the relationality of this work, and to inspire action on this important topic.

## 6. Limitations and Strengths

This study has a few limitations to note. First, this analysis represents only one listening session. In these types of sessions, although there is an opportunity for rich dialogue, there is also a likelihood that some members may suppress information or limit their contributions. This can be alleviated in future work by facilitating more than one listening session and making sure that the sessions are smaller so everyone has a chance to speak. A second limitation is that the sample was predominantly female and had higher levels of education than the general population of AI/AN people across the U.S., given that the data were gathered at a conference. This limitation likely means it is not generalizable to all AI/AN Elders, especially without the inclusion of more male-identifying participants. Further, this study used a convenience sample of individuals already in attendance at a conference focused specifically on Elders. Those in attendance at the conference were likely more mobile and willing to travel away from their home communities, which were largely rural- or reservation-based. Given NICOA's mission to advocate for Elders, these participants may already be aware of the health challenges facing this community.

Despite these limitations, there are many strengths to this project. First, there was a high number of participants that attended the event with the room being at full capacity. This high level of buy-in was a pleasant surprise to the authors and resulted in a larger-than-planned listening session, which could be mitigated by smaller sessions in the future. This may be due to the inclusion of a national, trusted partner that is committed and focused specifically on advocating for AI/AN Elders. These exploratory results may not be generalizable; however, they come from a unique sample that is often understudied. These Elders were not only willing to participate in this study but provided rich context and insights to the lead author as he continues to work with and alongside Elders.

**Author Contributions:** Conceptualization, C.A.; methodology, C.A.; formal analysis, C.A. and M.B.; investigation, C.A.; writing—original draft preparation, C.A.; writing—review and editing, M.B.; project administration, C.A.; funding acquisition, C.A. All authors have read and agreed to the published version of the manuscript.

**Funding:** This research received professional development funds from the Institute for Research to Educate and Advance Community Health (IREACH) at Washington State University to support analysis and compensation for participants.

**Institutional Review Board Statement:** The study was conducted in accordance with the Declaration of Helsinki and approved by the Institutional Review Board of Washington State University (IRB #19003; 4 August 2021) and the University of North Dakota (IRB0005334; 26 September 2022).

**Informed Consent Statement:** Informed consent was obtained from all subjects involved in the study as part of the listening session protocol used at the beginning of the session.

**Data Availability Statement:** The transcripts presented in this article are not readily available because they come from Tribal Citizens and are not readily available for public use. Requests to access the datasets should be directed to the lead author.

**Acknowledgments:** We want to thank the leadership and our colleagues at the Institute for Research to Educate and Advance Community Health (IREACH) at Washington State University for supporting this work.

**Conflicts of Interest:** The authors declare no conflicts of interest. The funders had no role in the design of the study; in the collection, analyses, or interpretation of data; in the writing of the manuscript; or in the decision to publish the results.

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
