# Peer review of "Visiting with Elders—Aging, Caregiving, and Planning for Future Generations of American Indians and Alaska Natives"

_genealogy, doi:10.3390/genealogy8020036_

Round 1

Reviewer 1 Report

Comments and Suggestions for Authors

Indigenous research methodologies look first to those impacted by colonisation to understand the particular situation being studied. Those knowledge holders will also know what the solutions are. So, to understand and address issues impacting the American Indian and Alaska Native Elders today, in this case Alzheimer's disease and related dementia including mild cognitive impairment, ask the Elders. That the literature in this area is silent on such an approach indicates a serious gap and this research provides an important exemplar for how to approach the area in an appropriate and meaningful way. It also highlights the urgent need for further research to find solutions. This article is a well-considered and important contribution.

Reflecting on research being conducted by other Indigenous Peoples, this is an important area of concern for Māori in Aotearoa New Zealand and the work of Indigenous researchers, Professor Sir Richard Faull and Dr Makarena Dudley (Centre for Brain Research, University of Auckland) on dementia in Māori elders could be of comparative interest to your team.

Reviewer 2 Report

Comments and Suggestions for Authors

Thank you for asking me to review this manuscript, Visiting with Elders - Aging, Caregiving, and Planning for Future Generations of American Indians and Alaska Natives. This paper covers an important topic that requires more attention.

Overall, the paper is clearly written and easy to follow. Please see below for suggestions for improvement.

Introduction:

The sentence on lines 31-33 (These rapidly…) could use additional information for clarity. Is it saying that it’s the rapidly aging community that has asked for the need to understand these issues or is it the fact that the numbers are rising that is making this need or something else?

Regarding the sentence on lines 43-44 (Doing so…), I wonder if it’s more than effective research or if you want to define what you mean by effective? The sentences above talk about communities receiving benefits, which doesn’t seem to be captured in this sentence.

Materials and methods:

Thank you for the researchers and reflexivity description.

The title of the table is missing on line 117

Regarding the sentence on lines 128-130 (Leadership at…), I wasn’t sure what it meant that the author presented his graduate program?

It would be helpful to define inductive coding for your readers (lines 161-162).

It sounds like you conducted data analysis across your questions versus doing analysis question by question. Please clarify.

Results:

It would be good to mention somewhere that some of the quotes presented could fit under multiple categories. What did you do when you saw that?

Content on lines 168-170 should be deleted.

A table displaying demographic data for the 38 participants would be helpful.

I wonder why there are example quotes in the table separate from the quotes in the narrative below? Seems they should all be together unless there’s a reason for the separation. Some of the quotes are the same in the table and below – they don’t need to appear twice.

I’m not sure why the third quote under intergenerational trauma (lines 200-201) fits there. Seems more related to ADRD. Maybe explain why it’s there if it’s going to stay.

Discussion:

It was nice to see the results reflected through the socio-ecological model.

Also under the individual level, it seems that the theme and findings related to intergenerational solidarity would fit and be discussed here.

The level after community is titled societal in one place and system in one place.

Limitations and strengths:

Thank you for the limitations section. This helps to place your study in a broader perspective. It seems that in addition to more listening sessions, the sessions could be limited to a smaller number of people in order to receive more information. Were you surprised that so many people showed up? You could share that if it is so.

You share in the beginning that it is important that data collected from people provides benefit. What did the authors do or are they doing to share back the information they learned with the participants or communities? How is this study going to provide benefit. The implications section is vague and broad.

Information is missing in lines 369-370.

Informed consent information is mentioned at the end. It would be good to include this in the body of the study as well. It seems difficult to gather informed consent if people are coming and going from the session. I’d like to see how this was done.

Comments on the Quality of English Language

minor edits needed
